Monitoring of UN sustainable development goal SDG-9.1.1: study of Algerian “Belt and Road” expressways constructed by China

Jia Zhanhai 1
Wu Mingquan wumq@radi.ac.cn 2
Niu Zheng 2
Tang Bin 1
Mu Yuxuan 3
1 College of Earth Sciences, Chengdu University of Technology , Chengdu , Si chuan Province , China
2 State Key Laboratory of Remote Sensing Science, Aerospace Information Research Institute, Chinese Academy of Sciences , Beijing , China
3 College of Tourism and Urban-Rural Planning, Chengdu University of Technology , Chengdu , Si chuan Province , China
Gavrilescu Maria
Electronic publication date: 2020 Jun 2
Publication date: 2020
Volume: 8
Electronic Location ID: e8953
Received 2019 Oct 3; Accepted 2020 Mar 21
Copyright: ©2020 Jia et al.
Copyright year: 2020
Copyright holder: Jia et al.
License: This is an open access article distributed under the terms of the Creative Commons Attribution License, which permits unrestricted use, distribution, reproduction and adaptation in any medium and for any purpose provided that it is properly attributed. For attribution, the original author(s), title, publication source (PeerJ) and either DOI or URL of the article must be cited.
License URL: https://creativecommons.org/licenses/by/4.0/

Keywords: SDG, Visual interpretation, NDBI, Estimation method

Funding: Strategic Priority Research Program of Chinese Academy of Sciences XDA19030304 Youth Innovation Promotion Association 2017089 This work was supported by the Strategic Priority Research Program of Chinese Academy of Sciences (No. XDA19030304) and the Youth Innovation Promotion Association CAS (No. 2017089). The funders had no role in study design, data collection and analysis, decision to publish, or preparation of the manuscript.

==============================
The proportion of the rural population who live within 2 km of an all-season road is an indicator of the United Nations’ Sustainable Development Goals (SDGs) 9.1.1. This paper aims to calculate SDG indicator 9.1.1 in the proximity of five Algerian expressways. Three monitoring methods are proposed for different spatial regions based on the five expressways built by China’s Belt and Road Initiative Project. These methods are based on remote sensing and WorldPop and The High Resolution Settlement Layer (HRSL) population data. The results indicate that (1) the WorldPop population statistics show that the five expressways built by China’s Belt Project have increased the rural population of the 2 km buffer zone by 192,016 between the start of construction and eight years after its completion. By the end of 2019, the population increased by 329,291 accounting for 1.17% of the rural population. (2) Based on populations estimated form built-up index (NDBI) building areas, the rural populations within the 2 km buffer area of the Bejaia-Haniff Expressway in 2011, 2015, and 2019 were 273,118, 306,430, and 375,408, respectively. (3) HRSL population grid statistics indicate that, in 2015, the populations were: East-West Expressway = 911,549, Bejaia Expressway = 127,471, Tipaza Expressway = 71,411, North-South Expressway = 30,583, and Cherchell Ring Expressway = 41,657. (4) A visual interpretation method based on Google Earth imagery was used to count the number of buildings and number of building floors in the town of Tikhramtath. Based on the estimated population of each building and floor, the population of Tikhramtath town in 2011, 2015, 2017, and 2019 was estimated as 1,790, 2,785, 3,365, and 3,870, respectively. (5) Through analysis and accuracy assessment, the appropriate statistical methods for different regions were determined.

Introduction

The United Nations (UN) Sustainable Development Goals (SDGs) were formally adopted at the UN Sustainable Development Summit in 2015. The goals aim to take 15 years to solve social, economic and environmental development issues in three dimensions, and adopt a sustainable development path. The SDGs contain 17 universal goals and 169 target objectives (Chen et al., 2018). An SDG indicator framework was officially announced and implemented in 2017 by the UN General Assembly. Since nine of these indicators are duplicated and belong to two or more targets, the actual index contains 232 items (Chen et al., 2018). Indicator 9.1.1 calculates the proportion of the rural population living within 2 km of an all-season road (Commission UNS, 2017).

Well-connected road transportation is an important part of achieving inclusive and sustainable growth. It can ensure that various production activities are carried out in an orderly manner so that each production factor can achieve a reasonable allocation of time and space (Bhattacharyay, 2015). Rural roads are the basic component of expressway networks and have important strategic significance in promoting regional economic development, improving the living standards of rural residents, and improving rural consumption. Indicator 9.1.1 focuses on assessing the level of transportation infrastructure construction and its impacts on economic development and people’s well-being (Xu, Bai & Chen, 2019). In 2016, the Chinese government issued the China Implementation Plan for the 2030 Agenda for Sustainable Development, which details China’s plans to implement 17 SDG universal goals and 169 target objectives within the next 15 years (Wei et al., 2018). However, at present, Chinese scholars have mainly conducted research and evaluation of domestic pilot areas, and research on foreign areas is lacking. China’s Belt and Road infrastructure interoperability major projects are fast, which has strongly promoted local socioeconomic development and population growth (Li et al., 2019; Chen & Li, 2018). The present study uses five expressways built in Algeria by China’s Belt and Road Initiative Project as an example with which to calculate progress in indicator 9.1.1.

At present, researchers in China and elsewhere have used several methods to evaluate the SDG-9.1.1 indicator. The World Bank Group defines the rural access index (RAI) as the proportion of the rural population who live within 2 km of an all-season road. This was defined together with an official method of measurement based on locally representative household surveys. Current estimates of the RAI indicate that some 900 million rural dwellers worldwide do not have adequate access to a formal transport system (Roberts, Kc & Rastogi, 2006). Mikou et al. (2019) used WorldPop population and Open Street Map network data to estimate the RAI and found that, assuming optimistic GDP growth, rural accessibility would only increase from 39 to 52 percent by 2030 across all developing countries. For this indicator, Mariathasan, Bezuidenhoudt & Olympio (2019) used Namibia as an example to calculate the possibility of road traffic through the global population grid. They compared four population datasets and the road networks of the OSM and NSA National Space Infrastructure (NSDI), and used the GEE APP tool to calculate the RAI. Xu, Bai & Chen (2019) added the three indicators road density, accessibility, and total postal service to build a new index. The revised RAI fully considers urban disadvantaged groups and eliminates the dependence of the original indicators on urban-rural boundary data. Qiu et al. (2019) used Deqing County in China as an example for evaluating the SDG-9.1.1 indicator. They used the building area and number of floors as weighting factors to establish a classification model. Population data with a spatial resolution of 30 m was obtained for sustainability analysis. SDG evaluations based on population data are more accurate and effective than those based on traditional methods. The Office for National Statistics (ONS) used the Ordnance Survey (OS) Master Map Integrated Transport Network (ITN) Layer as a basis for calculations, together with a population grid based on the 2011 Census, and the ONS’ urban/rural classification to select rural areas. The data showed that more than 99% of people live within 2 km of an all-season road (Frensis, 2018). The above research found that WorldPop population data is updated slowly and has low spatial resolution (the raster cell size is 100 m), making it impossible to count individual buildings. Most SDG-9.1.1 indicator evaluations are conducted at the national and provincial levels, and there is no precise way to evaluate small areas or newly constructed single expressway projects.

In response to this problem, we selected high-resolution remote sensing images and different population data. High-resolution remote sensing images have the advantages of high spatial resolution that provides rich and clear spatial information on ground features. It can clearly express spatial and surface texture structures and provide clear edge information and high temporal resolution (Ming et al., 2005). This paper takes five Algerian expressways built by China’s Belt and Road Initiative Project as an example. We propose three different methods for assessing indicator 9.1.1. The specific method objectives include: (1) Based on a large regional scope, an evaluation method based on HRSL population data is proposed to evaluate and analyse the contribution of five Algerian expressways built by China to the SDG-9.1.1 indicator in 2015. (2) Based on a medium-area scope, the NDBI building area population estimation method is used to evaluate and analyse the rural population that resided within 2 km of a road in 2011, 2015 and 2019 near the Bejaia-Hanif Expressway in Algeria. (3) Based on high-resolution remote sensing monitoring, the population of Tikhramtath town, which is within 2 km of the Bejaia-Hanif Expressway in Algeria, was assessed and analysed.

Research area and data

Research area

As part of China’s Belt Project, five expressways have been built in Algeria (Table 1). The total length of the East-West Expressway in Algeria is 1,216 km, of which 528 km of the middle and western sections are under construction by China International Trust Co., Ltd. and China Railway Construction Co., Ltd. The project has adopted the engineering procurement construction contracting mode and European technical standards. The Tipaza Expressway is 48 km long and was built by the China Construction Fifth Engineering Bureau Co., Ltd. It will promote the development of tourism in the Tipaza area. The North-South Expressway is 3,000 km long, and the China Construction Company undertook the 53 km Chiffa to Berrouaghia section. It is the most difficult and complicated section of the North-South Expressway, containing nearly 100 bridges and 76 viaducts. The project will strategically connect Algeria with Niger, Nigeria and Chad. The full length of the Cherchell Ring Expressway is 17 km, and it was built by the China Construction Company. The construction of the expressway greatly eased pressure on the N11 route and served as an extension of the 48 km expressway. The Bejaia Expressway is 100 km in length and was contracted by China Railway Construction. The connecting expressway project starts from the port of Bejaia in the north and connects to the East-West Expressway in the south. The project provides greater service to the port of Bejaia, while at the same time improving the local economy. The development has also eased traffic congestion in the area (Fig. 1). The construction of the expressway has promoted the growth of Eastern and Western economies in Algeria (Tian & Li, 2019), reduced the rate of traffic accidents, and formed a new space for social development (Bröcker, Dohse & Rietveld, 2019).

Table 1 China’s Belt and Road Expressway in Algeria.

Continent	Country	Highway name	Chinese company	Line length (km)	Construction start year	Construction end year or duration	Mode of cooperation (investment, construction, acquisition)	
Africa	Algeria	East-West Expressway	China Railway Construction 17th Bureau, China International Trust, and Investment	528	2006	2012	Joint construction	
Africa	Algeria	Tipaza Expressway	China Construction Fifth Bureau	48	2008	2011	Contract construction	
Africa	Algeria	North-South Expressway	China Construction Corporation	53	2012	2015	Joint construction	
Africa	Algeria	Cherchell Ring Expressway	China Construction Corporation	17	2014	2018	Contract construction	
Africa	Algeria	Bejaia Expressway	China Construction Corporation	100	2013	2017	Contract construction	

Research data

This paper uses WorldPop and HRSL population raster data, Landsat remote sensing imagery, Algerian urban area and administrative division map. The WorldPop project was launched in 2013 as part of a global population mapping project led by the Institute of Geographical Data, University of Southampton, UK (https://www.worldpop.org/). It aims to provide an open access archive of spatial demographic datasets for Central and South America, Africa and Asia to support development, disaster response and health applications. It uses raster prediction based on a random forest model to generate a 100 m spatial resolution population density, which is then used as a weighted surface and redistributed according to national population data (Stevens et al., 2015). LandScan uses the National Imagery and Mapping Agency’s (NIMA) Vector Map (VMAP) series data and Digital Terrain Elevation Data (DTED) for global coverage of roads and slopes, respectively (Bhaduri et al., 2002). HRSL population data has a resolution of 30 m and was obtained from the Center for International Earth Science Information Network (CIESIN) (Batran et al., 2018). Global Administrative Areas (GADM) is a database of the locations of the world’s administrative areas. Administrative areas in this database include countries, counties and departments (Hijmans, Garcia & Wieczorek, 2010). Landsat 5 and Landsat 8 data were obtained from the US Geological Survey (Table 2; El-Askary et al., 2014). These data were processed using several commands in ArcGIS 10.5 software.

Research methods

This paper aims to calculate the SDG 9.1.1 indicator in the proximity of five Algerian expressways built as part of China’s Belt and Road Initiative Project, from large to small areas. Figure 2 presents a flowchart of the three methods. (1) Obtaining large-scale statistics based on HRSL population data, using the nearest neighbor method and regional statistical method to evaluate and analyse the contribution of five Algerian expressways to monitor indicator 9.1.1 in the past five years; (2) For medium-sized areas, an evaluation method based on the NDBI building area index is proposed. (3) For small areas, an evaluation method based on high-resolution time series images is proposed. (4) analysis and comparison of the three population statistics methods.

Figure 1 Study area location maps.

(A) Tipaza Expressway, (B) Cherchell Ring Expressway, (C) North-South Expressway, (D) all five highways, (E) East-West Expressway, (F) Bejaia Expressway.

Table 2 Data sources.

Data	Source	Dates	Internet link	Spatial resolution	
World population raster data	HRSL	2015	https://www.ciesin.columbia.edu/data/hrsl/#data	30 m	
World population raster data	WorldPop	2010-2019	http://www.worldpop.org.uk/	100 m	
World population raster data	GHS	2015	https://ghslsys.jrc.ec.europa.eu/datasets.php#2016public	250 m	
Urban area	Natura Earth	2018	http://www.naturalearthdata.com	1:10 m	
Remote sensing image	Landsat 5 and Landsat 8	2011, 2015, 2019	http://earthexplorer.usgs.gov	30 m	
Administrative division map	GADM	2018	https://gadm.org/download_country_v3.html	Level-City	

Figure 2 Flow chart of the three methods.

(1) A large-scale evaluation method for indicator 9.1.1. based on HRSL population raster data.

In this paper, the population statistics of residents within two kilometers of five expressways are studied using HRSL population data and nearest neighbor and regional statistics methods. The nearest neighbor method locates the positions of pixels in the output image to the original image by the nearest neighbor method. It finds the nearest pixel from the original image and takes the value of the pixel as the value of the output image pixel. The nearest neighbor method is used to calculate the size of a single grid pixel. Then, by using the function of regional statistical analysis, all the grid pixel values are tabulated and, finally, the sum of all pixel values is calculated (Caraway, McCreight & Rajagopalan, 2014; Rossi, Dungan & Beck, 1994).

(2) Evaluation method of a medium area based on the NDBI building area index.

The Middle Area comprises the Bejaia-Haniff Link Expressway in Algeria and uses the NDBI building area population estimation method for demographic analysis. A two-kilometer NDBI distribution map of the Bejaia-Haniff Link Expressway road was calculated using Landsat remote sensing images from 2011, 2015, and 2019 and is presented in Fig. 3. The normalized difference building index (NDBI) indicates the distribution of land for regional construction (Zha, Ni & Yang, 2003; Li et al., 2017; Xu, 2008). The calculation method is shown in Eq. (1). The NDBI is mainly used to extract information on buildings in cities and towns but, according to Jin et al.’s (2020) research, it can also be used for villages and towns. There was a significant positive correlation between the remote sensing index of village buildings and surface temperature (P < 0.05). The Bejaia-Haniff Link Expressway has less urban area within 2 km. Most of the buildings are built in villages and towns. The article uses the city vector mask to remove the extracted urban buildings and retain the villages and towns.

Figure 3 2011, 2015, 2019 NDBI.

(A) Schematic diagram of the local study area, (B) 2011 Landsat remote sensing image map, (C) 2011 NDBI map, (D) 2015 Landsat remote sensing image map, (E) 2015 NDBI map, (F) 2019 Landsat remote sensing image map, (G) 2019 NDBI map.

(1) NDBI=SMIR−NIRSMIR+NIR

Where NDBI represents the normalized building index, SMIR represents mid-infrared reflectance, and NIR represents near-infrared reflectance.

Supervised classification: The maximum likelihood method is used to extract a sample from the whole population to ensure that the same block with the same band value as the sample is assigned the same attributes belonging to the same type of feature (Cabral et al., 2018; Murthy, Raju & Badrinath, 2003; Otukei & Blaschke, 2010; Keuchel et al., 2003). Combining remote sensing images of the study area with data on its land types, we categorized the land types as residential, forest, cultivated, and bare by visual interpretation. Then, 100 uniformly distributed samples were selected from the entire image (Fig. 4). After many experiments, the separability between each sample type was greater than 1.8 and the maximum likelihood algorithm was selected for classification. Finally, combined with 0.3m high-resolution remote sensing image for accuracy verification, the accuracy of the verification results is above 90% (Li et al., 2014; Zhang et al., 2016; Pujiono et al., 2019).

Figure 4 Supervised classification sample.

(A) Urben and rural areas sample. (B) Urban and rural verification images. (C) Woodland sample. (D) Woodland verification images. (E) Arable land sample. (F) Arable land verification images. (G) Bare land sample. (H) Bare land verification images.

NDBI was calculated using the Landsat-5 and Landsat-8 remote sensing image bands to extract the area of rural buildings, and the area of rural buildings within two kilometers of the expressway in 2011, 2015 and 2019 was estimated. In order to maintain the accuracy of extracting buildings and eliminate the effects of bare soil, cultivated land and water bodies, the bare soil and cultivated land were extracted by the supervised classification maximum likelihood method, while water bodies were extracted by visual interpretation. Finally, the NDBI was masked by areas of bare soil, cultivated land and water bodies to obtain a map of the rural building distribution. The total population is then calculated based on the total area of the building and the per capita living area.

(3) Small-scale evaluation method based on high-resolution time series images

The small area of the town of Tikhramtath in the Bejaia-Haniff buffer zone of Algeria was analysed in terms of demographics. A remote sensing method was used to monitor changes in buildings in 2011, 2015, 2017 and 2019 in small areas according to a multi-time remote sensing data map. A visual interpretation of the number of buildings and number of floors in the domains was carried out through Google Earth 0.24m high-resolution remote sensing images. First, the building area was calculated with the Calculate Geometry tool. Then, according to the size of the building, the population of each floor of each building was estimated to calculate the total population of the area.

(4) Accuracy evaluation and comparative analysis

Comparisons of the results of the three methods presented in the article were made. The root mean square error (RMSE) and relative RMSE (%RMSE) were used to measure the accuracy of WorldPop data, HRSL data, NDBI estimated building area data, and remote sensing visual interpretation data. The %RMSE values were obtained by dividing the RMSEs by the average of the number of censuses, which can reflect the accuracy of the model simulation. Finally, the most suitable population calculation methods for different regions were selected (Tan et al., 2017; Draper et al., 2013; Bhunia, Shit & Maiti, 2018).

(2) RMSE=1N∑fi−ri2

(3) % RMSE=RMSE1N ∑ri

Where fi is the estimated value of the ith group of data, that is, the estimated population density obtained after population spatialization; ri is the reference value of the ith data, that is, the population density value obtained from the census data; N represents group data.

Experimental results

Based on WorldPop population statistics

Algerian urban demographic results.

The statistics of the Algerian urban population are crucial to the statistics of the rural population. The rural population size is based on the total population minus the urban population. The distribution of the Algerian urban population from 2009 to 2019 is shown in Fig. 5. The sizes of the urban population in 2009, 2011, 2013, 2015, 2017 and 2019 were 11,396,261, 11,857,311, 12,632,705, 13,302,082, 14,019,482, and 14,820,436, respectively. In these years, the total rural population was 23,247,534, 24,126,163, 24,821,053, 25,788,594, 26,896,482, and 28,158,778 people.

Figure 5 Algerian urban population map.

(A) 2009, (B) 2011, (C) 2013, (D) 2015, (E) 2017, and (F) 2019.

According to the comparison of urban population changes, Fig. 5 shows that there are 84 cities in Algeria, five of which with an area of 200 km2 or more, all of which are located in the northern part of Algeria. The population of urban areas has grown rapidly over the last 10 years, with a total growth of 3,424,085 people. The urban areas along the new expressways are particularly obvious, including Oran, Alger, Blida, Bouira, Sidi-Bel-Abbes and other larger cities.

Figure 6 Demographic changes in the provinces of Algeria.

(A) Population in 2009–2011 and 2013, (B) Growth rate in 2011 and 2013.

Demographic results of the provinces of Algeria

Population data from the provinces of Algeria play an important role in calculating the proportion of the rural population that lives within 2 km of an expressway. Figure 6 shows that the top three most populous provinces in Algeria are Alger, Setif, and Oran, with populations of 3,037,455, 1,510,132 and 1,483,869, respectively. The three provinces with the smallest populations are Tindouf, Illizi, and Tamanghasset, with populations of 52,170, 54,636 and 181,217, respectively. The population growth rate of each province is low. In 2009-2011, the growth rate was above 10%, being 12.678% in Tindouf Province. In 2011–2013, the growth rate was above 10%.

The provinces with growth rates >10% during 2015–2017 were Naama Province, Tindouf Province, and Tiaret Province, with growth rates of 11.132%, 12.678% and 17.411%, respectively. Tizi Ouzou Province had the lowest growth rate at 0.516%. The provinces with growth rates >10% in 2017–2019 were the same, with growth rates of 11.694%, 12.678% and 22.306%, respectively, while Tizi Ouzou Province had the lowest growth rate at 0.545% (Fig. 7).

Figure 7 Demographic changes in the provinces of Algeria.

(A) Population in 2015, 2017 and 2019, (B) Growth rate in 2017 and 2019.

China’s construction of five expressways in Algeria: SDG-9.1.1 assessment

The Worldpop population raster data was counted as follows. (1) A expressway 2-km buffer surface was created using tool “Feature to polygon” (2) Natural Earth urban area data was based on the tool “Extract by mask” to obtain the urban population raster data, then the spatial analyze tool “Raster calculator” was used to obtain the rural population raster map. (3) The generated buffer surface was obtained using the “Extract by mask” tool to obtain 2-km rural population raster data. (4) The Worldpop cell raster layer was converted to a vector point layer through the “From raster to point tool” and the grid cell values were assigned to the point layer. (5) The required information was extracted from the point layer to a data table, then the population was counted (Tewari & Manning, 2017; Holt et al., 2018).

The distribution of the rural population after China’s construction of the Algeria expressway is shown in Fig. 8. The population of the Algeria Cherchell Ring Expressway is gradually increasing as shown in Table 3. Before the construction in 2011-2013, the growth rate was 1.77%. The population increased rapidly after the start of construction and was 6.66% in 2015. Population growth during the construction period was slower, dropping to 3.03%, and then increased afterward. From the start of construction to two years later, there was an increase of 5,733 people. The population growth rate within 2 km of the 53 km North-South Expressway decreased by 0.22% in 2013. After the start of construction, it gradually increased in 2017. After construction, the growth will be slower. The 53 km north-south expressway in Algeria will increase by 1,715 people from the start of construction to the second year after construction. The growth rate of the Bejaia-Haniff connection line of Algeria from pre-construction in 2011 to construction in 2015 gradually increased to 3.29%, the growth rate in 2015-2017 fell to 2.48%, and the growth rate after construction was 2.65%. The population increased by 15,825 during the Bejaia-Haniff Expressway construction period.

Figure 8 Population distribution within the 2 km buffffer zone of five Algerian highways constructed by China.

(A) 2009, (B) 2011, (C) 2013, (D) 2015, (E) 2017, and (F) 2019.

Table 3 Statistics of highway populations.

Year	Cherchell Ring Expressway population	Growth rate	North-South Expressway population	Growth rate	Bejaia Expressway population	Growth rate	
2011 (before construction)	39,581	4.82%	35,168	10.45%	136,392	2.18%	
2013 (before construction)	40,282	1.77%	35,246	0.22%	140,084	2.71%	
2015 (under construction)	42,964	6.66%	35,681	1.23%	144,698	3.29%	
2017 (under construction)	44,265	3.03%	36,883	3.37%	148,286	2.48%	
2019 (after construction)	46,015	3.95%	37,950	2.89%	152,217	2.65%	

As shown in Table 4, the population near the Tipaza expressway decreased. After the project started, the growth rate increased rapidly to 5.84%. During the construction process, the population growth rate dropped to 5.79%. After construction, the growth rate reached 7.09% two years later in 2013. The growth rate dropped to 2.66% from 2013 to 2015, and the population increased by 11,988 from the start to the end of construction. The growth rate of the Algeria East-West Expressway gradually increased from 2005 to 2009 to 6.36%. During the construction process, the population growth rate dropped to 3.68%. After completion of construction, the population growth rate was 4.85%. The population increased by 156,755 people during the East-West Expressway construction period. China’s construction of five expressways in Algeria has increased the rural population within the 2 km buffer zone by 192,016 in the eight years of construction. By 2019, the population increased by 329,291, accounting for 1.17% of the rural population. In 2009, 2011, 2013, 2015, 2017, 2019, the rural population within the 2 km buffer zone accounted for 4.79%, 4.80%, 4.87%, 4.89%, 4.90%, and 4.86% of the total rural population, respectively.

Table 4 Statistics of the highway populations.

Year	Tipaza 48 km highway population	Growth rate	East-West Expressway population	Growth rate	
2005 (before construction)	60,320	3.70%	764,851	3.61%	
2007 (before construction)	60,212	−0.18%	797,051	4.21%	
2009 (under construction)	63,729	5.84%	847,763	6.36%	
2011 (under construction)	67,419	5.79%	878,946	3.68%	
2013 (after construction)	72,200	7.09%	921,606	4.85%	

Large area range HRSL population raster data statistics results

The large-area HRSL population raster data analysis method is used the nearest neighbor method to calculate the size of a single grid pixel. Then, by using the ArcGIS function of regional statistical analysis, all the HRSL population raster grid pixel values are tabulated and, finally, the sum of all pixel values is calculated.

The distribution of HRSL raster data for the rural population living within 2 km of an expressway in 2015 is shown in Fig. 9. The populations of the Algeria East-West Expressway, Bejaia Expressway, Tipaza Expressway, North-South Expressway, and Cherchell Ring Expressway were 911,549, 127,471, 71,411, 30,583 and 41,657, respectively, accounting for approximately 4.12% of the 2015 rural population.

Figure 9 Distribution of rural HRSL population data within 2 km of highways in 2015.

(A) Bejaia Expressway, (B) Tipaza Expressway, (C) Cherchell Ring Expressway, (D) North-South Expressway, (E) East-West Expressway.

The population living within 2 km of the East-West Expressway was the largest of the five expressways, with a total population of 1,208,199, as shown in Table 5. The population distribution was closely related to the geography of Algeria. The northern part of Algeria comprises the coastal plains and hills of the Mediterranean coast, the central part is the Tell Atlas and Sahara Atlas Mountains, and the southern part of the Sahara desert is largely uninhabited.

Table 5 HRSL population statistics within 2 km of highways in 2015.

Highway	Rural population	Urban population	Total population	
Cherchell Ring Road	41,657	0	41,657	
North-South Expressway	30,583	19,844	50,427	
Bejaia Expressway	127,471	19,136	146,607	
Tipaza 48 km highway	71,411	17,336	88,747	
East-West Expressway	911,549	296,650	1,208,199	

Monitoring results of population estimation method for NDBI building area in the middle area

Figure 10 shows that the building areas within 2 km of the Bejaia-Haniff Expressway in 2011, 2015 and 2019 were 15.84 km2, 17.77 km2, and 21.77km2, respectively in Algeria. By sampling 100 buildings to determine that the mean single building area is 290 m2, the rural populations in 2011, 2015 and 2019 are estimated to be 273,118, 306,430 and 375,408, respectively. They accounted for 15.23%, 16.65%, and 19.64% of the rural populations of Bejaia, Bouira, and Boumerdes provinces, respectively.

Figure 10 NDBI extraction of rural buildings within 2 km of the Bejaia Expressway.

(A) 2011, (B) 2015, and (C) 2019.

The rural population of the Bejaia-Haniff Expressway in Algeria had the largest population growth rate in 2019 of 22.51%. Among them, the Tahalacht Industrial Zone in the northern part of the expressway, the Beni Mansour and the Tikhramtath town in the southern region, the Tala lbir town in the central region, Sidi Elash and Sidi Ayad have increased significantly.

Estimated population by remote sensing monitoring of a small area

Tikhramtath town is located at the beginning of the Bejaia Expressway, the closest to the Bejaia Expressway, and the towns and buildings are concentrated. The buildings increased significantly in 2011, 2015, 2017 and 2019, making it particularly suitable for small-area case selection.

The size of the building was vectored as a polygon feature by ArcGIS, then the ArcGIS tool “Calculate geometry” was used to calculate the area of the building. According to the 0.24-m high-resolution multi-time-series remote sensing image map, the number of building layers was compared with the image data of different periods. The number of layers of a single building was determined from the remote sensing image map. Finally, the layers of all buildings in Tikhramtath town were counted (Saadaoui et al., 2019).

A small area containing the town of Tikhramtath, within 2 km of the Bejaia-Hanif Expressway in Algeria (before, during and after construction) was selected for demographic analysis. A map of the building distribution is shown in Fig. 11. The population of Tikhramtath town in 2011, 2015, 2017 and 2019 was 1,790, 2,785, 3,365 and 3,870, respectively, accounting for 0.29%, 0.35%, 0.52%, and 0.60% of the total rural population in Bouira.

Figure 11 Remote sensing monitoring of the building distribution in Tikhramtath town.

(A) 2011, (B) 2011 and 2015, (C) 2011, 2015 and 2017, and (D) 2011, 2015, 2017, and 2019.

From the statistical chart, the size of buildings in the town of Tikhramtath within 2 km of the Bejaia-Hanif Expressway, the floor height, and the population of each floor of each building were estimated. The population of the Tikhramtath town in 2015, 2017 and 2019 is shown in Table 6. The growth rates in 2015, 2017 and 2019 were 55.59%, 20.83%, and 15.01%, respectively. The increase in population was mainly due to the construction of the Bejaia-Haniff Expressway.

Discussion

Comparative analysis

(1) WorldPop and HRSL population statistics were used to compare regions around five expressways built by China’s Belt and Road Initiative Project in 2015. The results are shown in Table 7.

The table shows that there is a small difference between the WorldPop demographic data and the HRSL population data. The rural populations of the five expressways buffers differed by 1307, 5098, 17,227, 2706, and 51,172.

(2) Comparison of the World Bank’s WorldPop population statistics method, the NDBI building area estimation method, and the HRSL population statistics method for the 2 km population of the Bejaia-Hanif Expressway.

This paper uses the World Bank’s WorldPop population statistics, the NDBI building area estimation method, and HRSL population statistics to analyse the demographics of the Begaia-Haniff Expressway in Algeria. The results are shown in Table 8.

According to the statistics, the WorldPop population data and NDBI building area population estimates for 2011, 2015, and 2019 are quite different. The differences are 136,726, 161,732 and 223,191, respectively. The difference between the WorldPop and HRSL population statistics is small, at 17,227. The NDBI building area population estimation method is subjective, and the total population can only be judged roughly, resulting in large errors. Worldpop demographics and HRSL demographics are also due to the small difference in statistical results due to resolution issues.

(3) Table 9 compares the WorldPop population data and the small area remote sensing monitoring method for Tikhramtath town.

Table 6 Population of Tikhramtath town according to building characteristics, 2011–2019.

Building area (m2)	2011	2015	2017	2019	Population per floor	
	1-floor buildings	2-floor buildings	3-floor buildings	1-floor buildings	2-floor buildings	3-floor buildings	1-floor buildings	2-floor buildings	3-floor buildings	1-floor buildings	2-floor buildings	3-floor buildings		
0–150	18	0	0	31	0	0	37	0	0	42	0	0	5	
150–300	57	4	0	124	6	1	145	11	1	159	11	1	10	
300–600	28	3	0	34	3	0	42	3	0	58	3	0	15	
>600	17	2	2	22	2	2	28	2	2	33	2	2	20	
Total population	1,790	2,785	3,365	3,870		

Table 9 shows that the results of the three methods for Tikhramtath town are quite different. The remote sensing monitoring visual interpretation method and WorldPop population statistics differed by 1782, 2766, 3349 and 3853, respectively. The remote sensing monitoring visual interpretation method differed from the HRSL population statistics by 2,690 people. The WorldPop population statistics are obviously inconsistent with the actual situation, while the remote sensing monitoring visual interpretation method locates the added buildings.

Table 7 WorldPop and HRSL population statistics.

Region	WorldPop population statistics (persons)	HRSL population statistics (persons)	
Cherchell Ring Expressway	42,964	41,657	
North-South Expressway	35,681	30,583	
Bejaia Expressway	144,698	127,471	
Tipaza Expressway	74,117	71,411	
East-West Expressway	962,721	911,549	

Table 8 WorldPop, NDBI and HRSL population statistics.

Time	WorldPop population statistics	NDBI building area population estimation method	HRSL population statistics	
2011 (before construction)	136,392	273,118		
2015 (under construction)	144,698	306,430	127,471	
2019 (after construction)	152,217	375,408		

Table 9 WorldPop, Remote sensing, and HRSL population statistics.

Time	WorldPop population statistics	Remote sensing visual interpretation	HRSL population data statistics	
2011 (before construction)	8	1,790		
2015 (under construction)	19	2,785	95	
2017 (under construction)	16	3,365		
2019 (after construction)	17	3,870		

This paper aims to calculate SDG indicator 9.1.1 in the proximity of five Algerian expressways. The SDG-9.1.1 indicator represents the proportion of the rural population who live within 2 km of an all-season road. At present, researchers in China and elsewhere have used many methods to evaluate the SDG-9.1.1 indicator. Qiu et al. (2019), Mariathasan, Bezuidenhoudt & Olympio (2019) and Xu, Bai & Chen (2019) have proposed different methods. However, most of the above methods use existing population data that is updated slowly, such that the new population cannot be found in time. This paper proposes the use of remote sensing to monitor newly added buildings to estimate the new rural population. This method makes up for the problem of the slow update of existing population data.

Precision verification

We used GHS Population Grid data for accuracy verification. The GHS Population Grid is made up of residential population estimates for target year 2015 provided by CIESIN GPWv4 (Gridded Population of the World, now in its fourth version). The estimates were disaggregated from census or administrative units into grid cells, according to the distribution and density of built-up areas as mapped in the GHSL global layer for the corresponding epoch. An accuracy assessment table is shown in Table 10.

Table 10 Accuracy verification table.

Area	Expressway	Source	RMSE	%RMSE	
Large area range	Cherchell Ring Expressway	WorldPop	26,516	10.01	
	North-South Expressway	
	Bejaia Expressway	HRSL	50,270	18.98	
	Tipaza Expressway				
East-West Expressway	
Middle area	Bejaia Expressway	WorldPop	10,516	6.78	
		HRSL	27,743	17.87	
NDBI building area estimate	15,216	97.42	
Small area	Tikhramtath town	WorldPop	751	97.53	
		HRSL	675	87.66	
Remote sensing visual interpretation data	2015	261.69	

In the accuracy comparison, lower %RMSE values indicate higher accuracy. The accuracy of the WorldPop population data in the large area was 89.99%, which is slightly higher than the 81.02% accuracy of the HRSL population data. WorldPop population data was also most accurate in the middle region. The NDBI building area estimates are subjectively influenced by people and had the lowest accuracy of 2.58%. The accuracy of the population data for Tikhramtath town (in the small area) was generally low and that of the HRSL population data was higher than that of the other two groups, at 12.34%.

Conclusions

Figure 12 shows the rural population within 2 km of the five Algerian Expressways in 2009, 2011, 2013, 2015, 2017, and 2019. It clearly shows that the rural population is greatest near the East-West Expressway and lowest near the North-South Expressway. An abnormal point was found. After searching, it was found that the population in 2009 was 31,842, indicating The rural population growth rate was relatively fast in 2011. It was found that the longer the mileage of rural roads, the greater the rural population within 2 km of a road.

Figure 12 The rural population within 2 kilometers of five Algerian Expressways in 2009, 2011, 2013, 2015, 2017, and 2019.

(A) Cherchell Ring Expressway and North-South Expressway, (B) Tipaza Expressway and Bejaia Expressway, (C) East-West Expressway.

The five expressways built by China’s Belt and Road Initiative Project have led to increases in the rural populations on both sides of the roads. Cherchell Ring Expressway, the 53 km North-South Expressway, Tipaza Expressway, Bejaia-Haniff Link Expressway, and East-West Expressway have driven rural population increases of 5733, 1715, 11,988, 15,825 and 192,016, respectively. The population of the 2 km buffer zone of the Bejaia-Haniff Expressway in Algeria in 2011, 2015 and 2019 was 273,118, 306,430, and 375,408, respectively. The population of Tikhramtath in 2011, 2015, 2017 and 2019 was 1790, 2785, 3365, and 3,870, respectively.

From large areas to small areas, different methods are suitable for road-wide demographic analysis: large- and medium-area demographics are suitable for WorldPop population statistics; while small-area demographics can use HRSL population raster data. The accuracy of the WorldPop population data in the largest area was 89.99%, while that of the HRSL population data was 81.02%. WorldPop population data was also most accurate in the middle region. The NDBI building area estimates had the lowest accuracy of 2.58%. The HRSL population data was higher than that of the other two groups, at 12.34% in the small area.

The WorldPop dataset is updated annually. HRSL population data is only available from 2015 and it is not possible to obtain data on populations within 2 km of roads every year. Although the NDBI building area data and remote sensing visual interpretation data are of low precision, they are not affected by slow population data updating and can estimate the population according to remote sensing images in a timely manner.

Supplemental Information

Supplemental Information 1 2011 Landsat image

Click here for additional data file.

Supplemental Information 2 2014 Landsat image

Click here for additional data file.

Supplemental Information 3 2019 Landsat image

Click here for additional data file.

Supplemental Information 4 2011 Worldpop Population Image

Click here for additional data file.

Supplemental Information 5 2013 Worldpop Population Image

Click here for additional data file.

Supplemental Information 6 2015 Worldpop Population Image

Click here for additional data file.

Supplemental Information 7 2017 Worldpop Population Image

Click here for additional data file.

Supplemental Information 8 2019 Worldpop Population Image

Click here for additional data file.

Supplemental Information 9 2015 hrsl Population Image

Click here for additional data file.

Additional Information and Declarations

Competing Interests

Author Contributions

Data Availability

The authors declare there are no conflicts of interest.

Zhanhai Jia conceived and designed the experiments, performed the experiments, analyzed the data, prepared figures and/or tables, and approved the final draft.

Mingquan Wu conceived and designed the experiments, authored or reviewed drafts of the paper, and approved the final draft.

Zheng Niu and Bin Tang analyzed the data, authored or reviewed drafts of the paper, and approved the final draft.

Yuxuan Mu performed the experiments, prepared figures and/or tables, and approved the final draft.

The following information was supplied regarding data availability:

The raw data is available in the Supplemental Files.

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
