# Peer review of "Monitoring of UN sustainable development goal SDG-9.1.1: study of Algerian “Belt and Road” expressways constructed by China"

_PeerJ, doi:10.7717/peerj.8953_

## Round 0.1 · original submission · Major Revisions

The peer reviewers' reports contain both positive comments, and considerations and recommendations for paper improvement.
Therefore, the authors are invited to revise and improve the manuscript according to the suggestions and comments of all peer reviewers, including those from the annotated manuscript attached.

Considering the opinion of one of the peer reviewers addressed to the editor, more details on the statistical methods used to evaluate the different populations could add value to the article.

Also, a graph comparing the accuracy of the different methods could help to interpret the conclusions in an easier and clearer manner.

A more consistent literature review based on previous researches performed in this area would be opportune.

·

Basic reporting

I found the manuscript very unclear and ambiguous. The introduction didn't have enough and well-structured paragraphs, lack of consistency. I put all my comments in the PDF.
Figures and Tables were well illustrated, visible and clear. However, there are minor mistakes, please find my comments in the pdf attached.
English should be improved to ensure that an international audience can clearly understand your text. I provided some examples to improve, but it really needs to be fully revised by the authors.
I also found that the authors need to familiarise themselves with the following terms and their meanings: sustainable Development Goals, Targets, and Indicators. There was a lot of confusion in the text.
Many of the references need to be revised.

Experimental design

Even if the article fits within the aim and scope of the journal, unfortunately, the aim of this study, research questions were unclear and undefined.
A rigorous methodology was attempted, however, I found a major mistake in the NDBI calculation, therefore, major revisions need to be accomplished.

Validity of the findings

Please see my comments in the pdf attached. Thank you.

Additional comments

Please find attached the PDF where I left the comments. Please follow the suggestions. If you don't agree please state the reason. Best wishes, AA

Reviewer 2 ·

Basic reporting

The manuscript needs some further improved before to be accepted for publication. In general, there are still some occasional grammar errors through the manuscript especially the article ‘’the’’, ‘’a’’ and ‘’an’’ is missing in many places, please make a spellchecking in addition to these minor issues. The reviewer has listed some specific comments that might be helpful of the authors to enhance the quality of the manuscript further.

Experimental design

• This section is also well written.
• Please explain what was the data resolution used in this study (hourly, daily…) and why?
• Please explain, why did the authors have chosen a case study from Algeria to compare with those from China?
• Did the authors used the census data to validate the data retrieved from remote sensing, please explain?
• What was the raster resolution used in this study?

Validity of the findings

Some sensitivity analysis is needed, perhaps using box plots.
The discussion should provide a summary of the main finding(s) of the manuscript in the context of the broader scientific literature, as well as addressing any limitations of the study or results that conflict with other published work.
Please include some quantitative findings in the conclusion section.

Additional comments

In general the manuscript is well written and deserve to be published in PeerJ journal.

·

Basic reporting

1) In Figure 3, the Landsat images for 2011, 2015 and 2019 are shown. But, in the raw data, the Landsat images for 2011, 2014 and 2019 are provided.

2) In Figure 5, the population is shown on the X-axis. The values shown on the X-axis are so close to each other as they are high numbers. The log of the values can be taken, to make the bar chart more readable.

3) In Figure 5, only 36 provinces are shown. As mentioned in Wikipedia, https://en.wikipedia.org/wiki/Provinces_of_Algeria there are 48 provinces. Please mention if other provinces due to valid reasons.

4) Figure 5 can be enlarged to show the provinces mentioned in the lines between 224 and 231. For instance, the provinces Algiers and Oran are mentioned but those are not visible in the figure.

5) Across the articles, the Tipaza highway and Cherchell Ring Expressway are alternatively referenced. Please try to use the same name or mention the alias name in the article. Similarly, there is the usage of Bekaya highways and Bejaia Highways.

6) In Figure 1, the Tibza expressway is not highlighted in the picture.

7) Please maintain the same level of line spacing in Table 3.

8) In table 4, the header name has Muridale ring road, instead of the Cherchell Ring Expressway. Please mention the alias in the article if applicable.

9) In the Abstract section, the numbering as made (2), (3), (4) and (5). But the first number (1) is missing.

10) Across the article, the population values for each year are mentioned continuously which affects the readability.

For instance, in line 42, the population values are mentioned as 911,549, 127,471, 71,411, 30,583 and 41,657.

Instead, it can be mentioned as " HRSL population grid statistics indicate that the East-West expressway has 911,549, Bejaia Expressway has 127,471.......population count in 2015"

The values between lines 255 to 257 are very much clustered. Please consider placing them in a table or rewording them as mentioned above.

11) Please have similar wording for SDG9.1.1 as it is also represented as SDG-9.1.1, in lines 81, 87, 129 132 and 133.

12) There was no Literature Review explaining any previous researches performed in this area.

Experimental design

The article suits all of the criteria mentioned in this section.

Validity of the findings

In the conclusion section, between lines 414 and 416, the reason to select the respective methods, which is the accuracy from the RMSE value should be mentioned.

Additional comments

1) This a great work in gathering data from different sources and performing spatial analysis.

2) Try to avoid the clustering of numbers and make the articles more readable. It would help to identify the statistical values quickly from the article.

3) It's a nice analysis done using ArcGIS which is a sophisticated tool.

---

## Round 0.2 · accepted · Accept

Dear Authors,

Your manuscript has been accepted for publication. in PeerJ - Environmental Science Section. Congratulations!
You are invited to follow the instructions of the editorial staff in the stage of proof correction and publication.

Thank you for choosing PeerJ and for your kind cooperation with our journal.

Best regards,

Maria Gavrilescu

Reviewer 2 ·

Basic reporting

Accept!

Experimental design

Accept!

Validity of the findings

Accept!

Additional comments

Accept!

·

Basic reporting

The article is revised well with all suggested changes. The figures, tables and the content are corrected as recommended.

Experimental design

The article suits all of the criteria mentioned in this section.

Validity of the findings

The recommended changes are updated as expected.

Additional comments

Thanks for updating the article based on the review comments.